# Changes in the estimated glomerular filtration rate and predictors of the renal prognosis in Japanese patients with type 2 diabetes: A retrospective study during the 12 months after the initiation of tofogliflozin

**Hiroyuki Ito**⬥*, **Hideyuki Inoue, Takuma Izutsu, Suzuko Matsumoto, Shinichi Antoku, Tomoko Yamasaki, Toshiko Mori, Michiko Togane**

Department of Diabetes, Metabolism and Kidney Disease, Edogawa Hospital, Edogawa-Ku, Tokyo, Japan

* ito@edogawa.or.jp

## Abstract

### Background

The changes in the estimated glomerular filtration rate (eGFR) and predictors of the renal prognosis were retrospectively assessed over the 12 months after the initiation of tofogliflozin, which has the shortest half-life among sodium-glucose cotransporter 2 (SGLT2) inhibitors, in Japanese patients with type 2 diabetes and renal impairment.

### Methods

In total, 158 patients treated with tofogliflozin between 2019 and 2021 were studied as the safety analysis set. One hundred and thirty subjects whose medication was continued over 12 months were investigated as the full analysis set. The subjects were divided into two groups based on the eGFR: normal- (eGFR $\geq$60 mL/min/1.73 m$^2$, n = 87) and low- (eGFR <60 mL/min/1.73 m$^2$, n = 43) eGFR groups.

### Results

The body weight, blood pressure, urinary protein excretion, and serum uric acid concentration decreased from baseline in both eGFR groups while the hemoglobin level increased. The eGFR did not significantly differ over time, except for the initial dip (-4.3±9.6 mL/min/1.73 m$^2$ in the normal-eGFR group and -1.5±5.3 mL/min/1.73 m$^2$ in the low-eGFR group). The change in the eGFR at 12 months after the initiation of tofogliflozin was -1.9±9.0 mL/min/1.73 m$^2$ and 0.2±6.0 mL/min/1.73 m$^2$ in the normal- and low-eGFR group, respectively. In the normal-eGFR group, the change in the eGFR showed a significant negative correlation with the HbA1c and eGFR at baseline, according to a multiple regression analysis. In the low-eGFR group, the change in the eGFR showed a significant negative correlation with urate-lowering agent use. The frequencies of adverse events specific for SGLT2 inhibitors were not significantly different between the normal- and low-eGFR groups.

**Data Availability Statement:** All relevant data are within the paper and its Supporting Information file.

**Funding:** This work was partly supported by Kowa Company, Ltd. (https://www.kowa.co.jp/). The funder had no role in study design, data collection and analysis, decision to publish, or preparation of the manuscript. The funder provided the English editing and publication fees. There was no additional external funding received for this study.

**Competing interests:** H Ito has received funding support and lecture fees from Kowa Company, Ltd. and Taisho Pharmaceutical Co., Ltd., and lecture fees from Eli Lilly Japan KK, Novo Nordisk Pharma Ltd., Sumitomo Pharma Co., Ltd., Boehringer Ingelheim, Sanofi KK, Daiichi Sankyo Company, Novartis Pharma KK, Takeda Pharmaceutical Company Ltd., MSD KK, Astellas Pharma, Terumo Corporation, Mochida Pharmaceuticals, Teijin Pharma, Kissei Pharmaceuticals, Mitsubishi Tanabe Pharma Corporation, Sanwa Kagaku Kenkyusho, AstraZeneca KK, Kyowa Kirin Co. Ltd., Otsuka Pharmaceutical Co., Ltd., Bayer Yakuhin, Ltd., EA Pharma Co., Ltd., Ono Pharmaceutical Co., Ltd., Viatris Inc., and has received consulting fee from Becton, Dickinson and Company. H Inoue has received lecture fees from AstraZeneca KK. T Izutsu has received lecture fees from Boehringer Ingelheim, Novo Nordisk Pharma Ltd., Taisho Pharmaceutical Co., Ltd., Kowa Company, Ltd. and Asahi Kasei Pharma Corporation. S Matsumoto has received lecture fees from Eli Lilly Japan KK, Novo Nordisk Pharma Ltd., Astellas Pharma, Kyowa Kirin Co., Ltd. and AstraZeneca KK. S Antoku has received lecture fees from Kyowa Kirin Co. Ltd., Sanofi KK, Taisho Pharmaceutical Co., Ltd., Daiichi Sankyo Company, and Otsuka Pharmaceutical Co., Ltd. T Yamasaki, T Mori and M Togane have no conflict of interest. This does not alter our adherence to PLOS ONE policies on sharing data and materials.

## Conclusions

Tofogliflozin may preserve renal function in the medium term in patients with type 2 diabetes and kidney impairment without an increase in specific adverse events.

## Introduction

Sodium-glucose cotransporter 2 (SGLT2) inhibitors reportedly not only improve hyperglycemia but also suppress the onset of cardiovascular diseases and prevent renal dysfunction in patients with type 2 diabetes [1–5], including in Asian patients, whose body mass index (BMI) and estimated glomerular filtration rate (eGFR) are generally smaller than in Western people [6,7]. Although the renal function of subjects at the baseline has been mainly within the normal range in previous large clinical trials [2–4], recent reports have revealed that SGLT2 inhibitors exert renal protection in both diabetic and non-diabetic patients with renal impairment [8–10].

We also reported that the administration of the SGLT2 inhibitor luseogliflozin preserved the renal function and reduced the blood pressure, body weight and urinary protein excretion (uPE) in type 2 diabetic patients with an eGFR of <60 mL/min/1.73 m$^2$ without increased adverse events (AEs) specific for SGLT2 inhibitors [11]. In addition, we showed that empagliflozin improved the above metabolic factors and protected renal function similarly in both elderly and non-elderly patients with type 2 diabetes [12].

Despite having the shortest elimination half-life (5.29±0.508 h) of all SGLT2 inhibitors available in Japan [13,14], tofogliflozin administration has been reported not to be inferior with regard to improving blood glucose and HbA1c levels in patients with type 2 diabetes [14]. Furthermore, treatment with tofogliflozin is expected to preserve the patient's quality of life (QOL) by reducing the frequency of nocturnal urination [15,16], which is frequently seen in diabetic patients. Although renal protection by SGLT2 inhibitors seems to be a class effect, no studies using tofogliflozin have been conducted in patients with type 2 diabetes and renal impairment. It is considered important to determine whether tofogliflozin can exert renoprotective effects in actual clinical practice.

We therefore retrospectively investigated the changes in the eGFR and predictors of the renal prognosis in a sample of Japanese patients with type 2 diabetes who received tofogliflozin over 12 months. The primary aim of the study was to determine the difference in changes in the eGFR before and after the initiation of tofogliflozin. Second, we examined changes in the factors affecting the renal prognosis, such as body weight, blood pressure, uPE, HbA1c, hemoglobin and serum uric acid concentration, as well as the safety of tofogliflozin administration.

## Methods

### Study design and patients

This study was conducted under the initiative of the investigators. Fig 1 illustrates the flow chart for the patient selection process. Japanese adult outpatients with type 2 diabetes who visited Edogawa Hospital, Tokyo, Japan, for the treatment of diabetes, were extracted using the hospital computer. Two hundred and fifty Japanese patients with type 2 diabetes who received 20 mg of tofogliflozin once daily (Deberza® tablets; Kowa Company, Ltd., Nagoya Japan) at the Department of Diabetes, Metabolism and Kidney Disease from March 2019 to April 2021 were eligible for inclusion in this study. Subjects who had already been prescribed tofogliflozin at the initial consultation (n = 18) and subjects who stopped treatment or were transferred to other medical institutions during the observation period (n = 31) were excluded from the

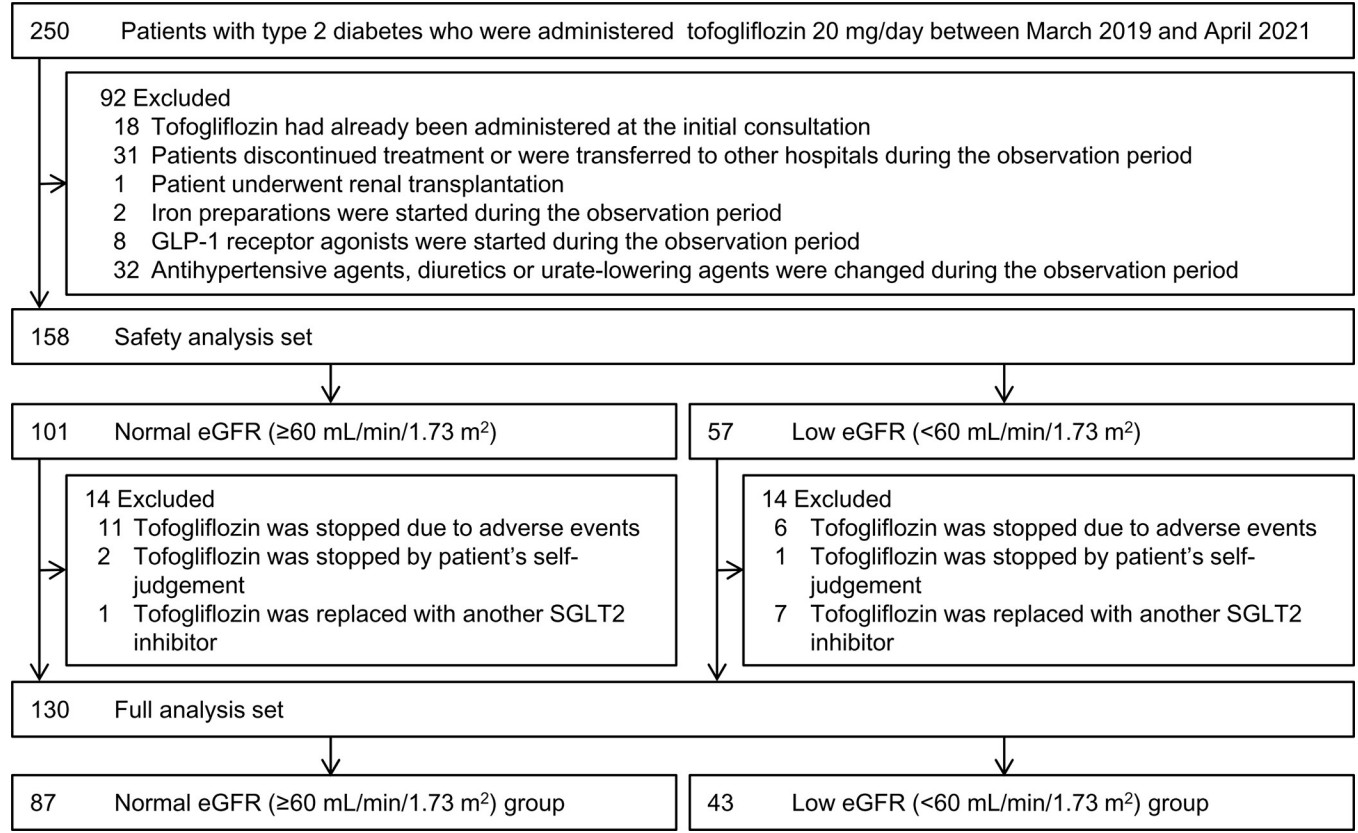

**Fig 1. Flowchart of patient selection.** The safety of tofogliflozin was analyzed in the safety analysis set (n = 158), and the effectiveness was investigated in the full analysis set (n = 130). The analysis sets were divided into the normal- ($\geq$60 mL/min/1.73 m$^2$, n = 87) and low- ($<$60 mL/min/1.73 m$^2$, n = 43) eGFR groups. GLP-1, glucagon-like peptide-1; eGFR, estimated glomerular filtration rate; SGLT2, Sodium-glucose cotransporter 2.

analysis. Subjects who had undergone renal transplantation (n = 1), those in whom iron preparation (n = 2) or GLP-1 receptor agonist treatment (n = 8) had been started during the observation period and those whose antihypertensive agents, diuretics or urate-lowering agent therapies were changed during the observation period (n = 32) were also excluded.

In total, 158 patients with type 2 diabetes (males: 71%, 64±13 years old) were studied as the safety analysis set (SAS) to analyze the safety of tofogliflozin. The SAS included 101 subjects with a normal eGFR ($\geq$60 mL/min/1.73 m$^2$) and 57 with a low eGFR ($<$60 mL/min/1.73 m$^2$). After excluding subjects who stopped tofogliflozin treatment due to any AEs (n = 17), stopped it based on their own judgment (n = 3) or had tofogliflozin replaced with another SGLT2 inhibitor (n = 8), 130 patients were investigated as the full analysis set (FAS) to determine the effectiveness of tofogliflozin.

Finally, the study subjects were divided into two groups based on the eGFR: normal-eGFR (n = 87) and low-eGFR (n = 43) groups. Moreover, the changes in the eGFR were investigated in 102 patients who had been treated in our department over 12 months before the initiation of tofogliflozin.

## Ethics conduct

The study was conducted in accordance with the principles expressed in the 2008 Declaration of Helsinki. The study protocol was approved by the Ethics Committee of Edogawa Hospital,

and written informed consent was waived as the data analyzed for this retrospective study were anonymous and based on information stored within the hospital (approval number: 2022–14, approval date: May 30, 2022). The trial is registered on UMIN-CTR, identifier UMIN000048277.

## Measurements

The patient backgrounds and laboratory findings were obtained from their medical records. The clinical parameters were measured similarly to our previous studies investigating the effects of SGLT2 inhibitors in patients with type 2 diabetes [11,12]. The eGFR was calculated based on the formula recommended by the Japanese Society of Nephrology [17]. The urinary albumin-to-creatinine ratio (uACR) and uPE were evaluated in a random spot urine test. Albuminuria was defined as a uACR of ≥30 mg/g/creatinine. The uPE was determined using urine test strips (Uriflet S; ARKRAY, Inc., Kyoto, Japan) and an automatic analyzer (Austin MAX AX 4280; ARKRAY, Inc.). Proteinuria was classified into four levels: (±), (1+), (2+) and (3+), which correspond to 15 mg/dL, 50 mg/dL, 150 mg/dL and 325 mg/dL, respectively. These levels are based on the median value of the measurement range in the semi-quantitative results [11,12,18,19]. Urinary liver-type fatty-acid binding protein (L-FABP), a reliable bio-marker for predicting renal dysfunction caused by tubulointerstitial injury in the kidneys in patients with diabetes [20–22], was measured using a chemiluminescent enzyme immunoassay at an external laboratory (SRL Co., Tokyo, Japan).

Obese subjects were defined as those with a BMI of ≥25.0 kg/m$^2$. Hypertension was defined as a systolic blood pressure of ≥140 mmHg and/or a diastolic blood pressure of ≥90 mmHg. Individuals currently using antihypertensive drugs were also classified as being positive for hypertension. Hyperuricemia was defined as a serum uric acid level of >327 μmol/L (7.0 mg/dL) or as patients using antihyperuricemics (benzbromarone, allopurinol, febuxostat or topir-oxostat). A current drinker was defined as a subject consuming >20 g ethanol equivalent/day. Diabetic retinopathy was graded based on the results of a fundus examination evaluated by ophthalmologists. Diabetic peripheral neuropathy was diagnosed by the presence of two or more related clinical symptoms, the absence of ankle tendon reflexes and reduced vibration sensations using a C128 tuning fork. Cerebrovascular disease was diagnosed as a history of ischemic stroke based on brain computed tomography or magnetic resonance imaging. Coronary heart disease was diagnosed as a history of myocardial infarction, angina pectoris or percutaneous coronary interventions.

The clinical parameters and AEs were retrospectively extracted over 12 months after the initiation of tofogliflozin from the subjects' medical records. If clinical data, including the body weight, blood pressure, uPE, HbA1c, hemoglobin and serum creatinine and uric acid concentration, were missing, we used the previously observed value following the last observational carried forward (LOCF) method.

## Statistical analyses

All data are presented as the mean±standard deviation. The $\chi^2$ test was used for between-group comparisons of categorical variables. Because none of the continuous variables (age, duration of diabetes, body weight, BMI, blood pressure, uACR, uPE, urinary L-FABP, HbA1c, hemoglobin, serum creatinine and uric acid and eGFR) showed a normal distribution in the Shapiro-Wilk tests, we utilized the Wilcoxon's signed rank test to determine the significance of differences in the continuous variables. Wilcoxon's rank sum test was used to assess the significance of differences in the body weight, blood pressure, uPE, uACR, urinary L-FABP, HbA1c, hemoglobin, serum uric acid and eGFR during the observation period compared to baseline

values. A linear least squares model was utilized to assess the correlations between the patient's clinical background factors, such as the duration of diabetes and comorbid diabetic vascular complications, and the changes in the eGFR at 12 months after the initiation of tofogliflozin therapy. A multivariate analysis was conducted to determine the significant factors that influenced changes in each dependent variable, including only those that showed a significant association in the earlier univariate analysis. P values of <0.05 (two-tailed) indicated statistical significance. The statistical software package JMP version 12.2.0 (SAS Institute, Cary, NC, USA) was used to perform all analyses. The sample size required to compare changes in the eGFR before and after the initiation of tofogliflozin was calculated using EZR version 1.42 (Saitama Medical Center, Jichi Medical University, Saitama, Japan), which is a graphical user interface for R (The R Foundation for Statistical Computing, Vienna, Austria).

## Results

### Baseline characteristics

**Table 1** presents the clinical characteristics of the FAS at the baseline. It was observed that the low-eGFR group had significantly older patients and a longer duration of diabetes compared to the normal-eGFR group. The prevalence of albuminuria and hyperuricemia were significantly higher in the low-eGFR group than in the normal-eGFR group. The use of antihypertensive drugs and insulin preparation and the values of uACR, uPE and urinary L-FABP were also significantly higher in the low-eGFR group than in the normal-eGFR group. The HbA1c and hemoglobin values did not differ markedly between the two groups.

### Changes in clinical parameters in the FAS

The changes in clinical parameters of the FAS from baseline to 12 months after the initiation of tofogliflozin are shown in **Table 2.** The body weight gradually decreased from the baseline value in both the normal- and low-eGFR groups. The systolic blood pressure significantly decreased at 1 month after the initiation of tofogliflozin and remained at the same level thereafter in both the normal- and low-eGFR groups. The diastolic blood pressure did not significantly differ in the normal-eGFR group between the time points. The changes in body weight, systolic blood pressure, and diastolic blood pressure did not significantly differ between the two groups (P = 0.78, P = 0.36, and P = 0.32, respectively). The uPE was halved at 1 month, and the subsequent decrease was mild in both the normal- and low-eGFR groups. The uACR and urinary L-FABP values significantly decreased in the low-eGFR group. The change in the uACR from baseline was significantly larger in the low-eGFR group than in the normal-eGFR group (P = 0.004). Although the changes in uPE and urinary L-FABP were larger in the low-eGFR group than in the normal GFR group, there were no significant differences (P = 0.12 and P = 0.14, respectively). The HbA1c was also reduced in both the normal- and low-eGFR groups after tofogliflozin administration, and the change in HbA1c tended to be smaller in the low-eGFR group than in the normal-eGFR group (P = 0.06). The hemoglobin level gradually increased from the baseline value in both the normal- and low-eGFR groups, and the change in the hemoglobin level was not significantly different between the two groups (P = 0.36). Although the serum uric acid concentrations were decreased in both groups, the reduction was significant only in the normal-eGFR group. The change in the serum uric acid concentration was not significantly different between the two groups (P = 0.65). The eGFR was significantly reduced from baseline at 1 month after the initiation of tofogliflozin and then gradually increased. The initial eGFR dips at 1 month were -3.4±8.5 mL/min/1.73 $m^2$, -4.3±9.6 mL/min/1.73 $m^2$ and -1.5±5.3 mL/min/1.73 $m^2$ in all subjects and the normal- and low-eGFR groups, respectively. The change in the eGFR at 12 months after the initiation of tofogliflozin was -1.9

**Table 1. The clinical characteristics of the full analysis set at baseline.**

| | N¶ | All subjects | Groups according to eGFR at baseline | | P |
| --- | --- | --- | --- | --- | --- |
| | | | Normal eGFR | Low eGFR | |
| | | | (n = 87) | (n = 43) | |
| Male sex (%) | 130 | 72 | 71 | 74 | 0.71 |
| Age (years) | 130 | 64±12 | 61±12 | 70±10 | <0.01 |
| Duration of diabetes (years) | 120 | 13±9 | 11±8 | 15±10 | 0.03 |
| Smoking history (%) | 128 | 57 | 54 | 63 | 0.35 |
| Current drinker (%) | 128 | 28 | 32 | 21 | 0.20 |
| Body weight (kg) | 124 | 74.0±14.2 | 74.0±14.8 | 73.9±13.1 | 0.85 |
| Body mass index (kg/m$^2$) | 124 | 27.6±4.8 | 27.6±5.1 | 27.7±4.4 | 0.78 |
| Systolic blood pressure (mmHg) | 122 | 135±15 | 134±12 | 139±18 | 0.23 |
| Diastolic blood pressure (mmHg) | 119 | 80±13 | 80±13 | 79±16 | 0.44 |
| Diabetic retinopathy (%)† | 110 | 29 | 24 | 39 | 0.11 |
| Diabetic peripheral neuropathy (%) | 101 | 33 | 29 | 39 | 0.32 |
| Cerebrovascular disease (%) | 130 | 17 | 15 | 21 | 0.39 |
| Coronary heart disease (%) | 130 | 18 | 16 | 23 | 0.32 |
| Obesity (%) | 124 | 69 | 66 | 74 | 0.37 |
| Hypertension (%) | 129 | 81 | 77 | 91 | 0.05 |
| Albuminuria (%) | 121 | 45 | 36 | 65 | <0.01 |
| Hyperuricemia (%) | 118 | 31 | 21 | 50 | <0.01 |
| RAAS inhibitors use (%)‡ | 130 | 52 | 46 | 65 | 0.04 |
| Calcium channel blockers use (%) | 130 | 57 | 34 | 60 | <0.01 |
| Urate-lowering agents use (%) | 130 | 25 | 15 | 44 | <0.01 |
| Metformin use (%) | 130 | 56 | 62 | 44 | 0.05 |
| Sulfonylureas use (%) | 130 | 6 | 6 | 7 | 0.78 |
| Thiazolidinediones use (%) | 130 | 5 | 6 | 2 | 0.38 |
| α-glucosidase inhibitors use (%) | 130 | 5 | 7 | 0 | 0.08 |
| Glinides use (%) | 130 | 2 | 3 | 0 | 0.22 |
| DPP-4 inhibitors use (%) | 130 | 42 | 38 | 51 | 0.15 |
| GLP-1 receptor agonists use (%) | 130 | 12 | 11 | 12 | 0.98 |
| Insulin use (%) | 130 | 20 | 15 | 30 | 0.04 |
| uPE (mg/dL) | 121 | 35.5±76.9 | 20.5±49.2 | 65.8±108.8 | <0.01 |
| uACR (mg/gCr) | 101 | 224.8±644.2 | 93.3±273.9 | 535.8±1052.5 | <0.01 |
| Urinary L-FABP (µg/gCr) | 58 | 12.0±26.0 | 9.7±28.9 | 16.6±18.9 | <0.01 |
| HbA1c (%) | 129 | 8.4±1.5 | 8.4±1.6 | 8.3±1.4 | 0.83 |
| HbA1c (mmol/mol) | 129 | 67.9±16.5 | 68.2±17.0 | 67.1±15.5 | 0.87 |
| Hemoglobin (g/L) | 130 | 143±15 | 144±14 | 142±15 | 0.62 |
| Uric acid (µmol/L) | 118 | 305±70 | 297±71 | 319±67 | 0.12 |
| eGFR (mL/min/1.73 m$^2$) | 130 | 68.9±21.8 | 80.7±15.3 | 44.6±10.4 | <0.01 |
| Minimum, maximum | | 23.1, 141.8 | 60.3, 141.8 | 23.1, 59.9 | |
| 25th percentile, median, 75th percentile | | 52.3, 70.0, 80.8 | 69.9, 77.3, 85.9 | 38.3, 46.2, 52.4 | |

eGFR, estimated glomerular filtration rate; RAAS, renin-angiotensin-aldosterone system; DPP-4, dipeptidyl peptidase-4; GLP-1, glucagon-like peptide-1; uPE, urinary protein excretion; uACR, urinary albumin-to-creatinine ratio; L-FABP, liver-type fatty acid-binding protein.

N: Number estimated.

† Diabetic retinopathy includes simple, preproliferative and proliferative retinopathy.

‡ RAAS inhibitors include angiotensin-converting enzyme inhibitors, angiotensin II receptor blockers and aldosterone receptor antagonists.

**Table 2. Changes in clinical parameters of the full analysis set.**

| | Baseline | 1 month | 3 months | 6 months | 9 months | 12 months | Change from baseline |
|---|---|---|---|---|---|---|---|
| Body weight (kg) | | | | | | | |
| All subjects (n = 91) | 74.3±14.7 | 72.9±14.5** | 72.2±14.4** | 71.8±14.6** | 71.6±14.7** | 71.5±14.7** | -2.9±3.5 |
| Normal-eGFR (n = 63) | 73.8±15.3 | 72.5±15.3** | 71.8±15.0** | 71.4±14.9** | 71.2±15.1** | 71.1±15.3** | -2.8±3.8 |
| Low-eGFR (n = 28) | 75.4±13.3 | 73.9±12.8** | 73.0±12.9** | 72.8±14.0** | 72.5±14.0** | 72.4±13.5** | -3.0±2.7 |
| Systolic blood pressure (mmHg) | | | | | | | |
| All subjects (n = 122) | 135±15 | 131±13** | 131±13** | 132±15* | 132±13* | 131±14** | -5±14 |
| Normal-eGFR (n = 82) | 134±12 | 130±12* | 130±12* | 132±14 | 131±12 | 130±13* | -4±13 |
| Low-eGFR (n = 40) | 139±18 | 133±15* | 133±15* | 133±16* | 132±15* | 132±16* | -7±16 |
| Diastolic blood pressure (mmHg) | | | | | | | |
| All subjects (n = 119) | 80±13 | 78±11 | 78±13 | 78±12 | 78±13 | 79±12 | -2±12 |
| Normal-eGFR (n = 80) | 80±12 | 78±11 | 80±13 | 79±11 | 79±11 | 80±12 | -0±11 |
| Low-eGFR (n = 39) | 79±16 | 78±12 | 75±12* | 77±13 | 75±16** | 77±12* | -2±9 |
| uPE (mg/dL) | | | | | | | |
| All subjects (n = 121) | 35.5±76.9 | 18.6±47.6** | 14.9±42.8** | 13.1±40.7** | 14.0±40.9** | 11.8±37.1** | -23.6±58.7 |
| Normal-eGFR (n = 81) | 20.5±49.2 | 9.3±26.3**t22 | 6.4±19.9** | 8.0±37.5** | 9.2±40.4** | 6.4±24.2** | -14.1±35.3 |
| Low-eGFR (n = 40) | 65.8±108.8 | 37.5±70.8** | 32.1±66.1** | 23.4±45.3** | 23.8±40.7** | 22.9±53.4** | -42.9±86.5 |
| uACR (mg/gCr) | | | | | | | |
| All subjects (n = 101) | 224.8±644.2 | | | 171.0±48.5* | | 151.0±438.5** | -73.7±259.1 |
| Normal-eGFR (n = 71) | 93.3±273.9 | | | 87.6±312.0 | | 80.9±271.8 | -12.0±80.8 |
| Low-eGFR (n = 30) | 535.8±1052.5 | | | 368.4±705.1 | | 316.9±666.6** | -218.8±429.8## |
| Urinary L-FABP (μg/gCr) | | | | | | | |
| All subjects (n = 58) | 12.0±26.0 | | | 9.6±26.6* | | 9.8±26.5* | -2.1±9.1 |
| Normal-eGFR (n = 39) | 9.7±28.9 | | | 9.0±31.2 | | 9.1±31.2 | -0.6±6.0 |
| Low-eGFR (n = 19) | 16.6±18.9 | | | 10.7±13.2* | | 11.3±12.6* | -5.2±13.2 |
| HbA1c (%) | | | | | | | |
| All subjects (n = 129) | 8.4±1.5 | 7.8±1.2** | 7.6±1.1** | 7.4±0.9** | 7.4±0.8** | 7.4±1.0** | -0.9±1.3 |
| Normal-eGFR (n = 86) | 8.4±1.6 | 7.9±1.3** | 7.6±1.2** | 7.4±0.8** | 7.3±0.8** | 7.3±0.9** | -1.0±1.3 |
| Low-eGFR (n = 43) | 8.3±1.4 | 7.8±1.0** | 7.7±1.1** | 7.4±0.9** | 7.4±0.9** | 7.6±1.1** | -0.7±1.1# |
| Hemoglobin (g/L) | | | | | | | |
| All subjects (n = 130) | 143±15 | 145±15** | 146±15** | 146±17** | 146±18** | 147±16** | 4±9 |
| Normal-eGFR (n = 87) | 144±14 | 146±15** | 148±15** | 147±18** | 147±19** | 148±17** | 4±10 |
| Low-eGFR (n = 43) | 142±15 | 143±15 | 142±15 | 143±15 | 145±15* | 145±15* | 2±9 |
| Uric acid (μmol/L) | | | | | | | |
| All subjects (n = 118) | 305±70 | 295±70* | 290±66** | 290±68** | 287±64** | 286±63** | -18±50 |
| Normal-eGFR (n = 78) | 297±71 | 289±75 | 283±67* | 281±70** | 277±63** | 276±65** | -21±49 |
| Low-eGFR (n = 40) | 319±67 | 306±59 | 304±63 | 307±63 | 307±60 | 305±53 | -13±54 |
| eGFR (mL/min/1.73 m$^2$) | | | | | | | |
| All subjects (n = 130) | 68.7±21.8 | 65.3±21.9** | 66.7±21.1** | 67.7±21.6 | 67.6±20.6 | 67.5±21.1 | -1.2±8.2 |
| Normal-eGFR (n = 87) | 80.6±15.0 | 76.3±16.9** | 77.8±14.6** | 78.5±13.8* | 78.6±13.8 | 78.7±14.4 | -1.9±9.0 |
| Low-eGFR (n = 43) | 44.6±10.4 | 43.1±11.3** | 44.2±12.5 | 45.5±13.1 | 45.2±12.0 | 44.8±12.3 | 0.2±6.0 |

eGFR, estimated glomerular filtration rate; uPE, urinary protein excretion; uACR, urinary albumin-to-creatinine ratio; L-FABP, liver-type fatty acid-binding protein.

* P<0.05

** P<0.01 vs. corresponding value at baseline, # P <0.1, ## P<0.01 vs. corresponding value in the normal-eGFR group.

±9.0 mL/min/1.73 m$^2$ and 0.2±6.0 mL/min/1.73 m$^2$ in the normal- and low-eGFR groups, respectively. The change in the eGFR was not significantly different between the two groups (P = 0.44).

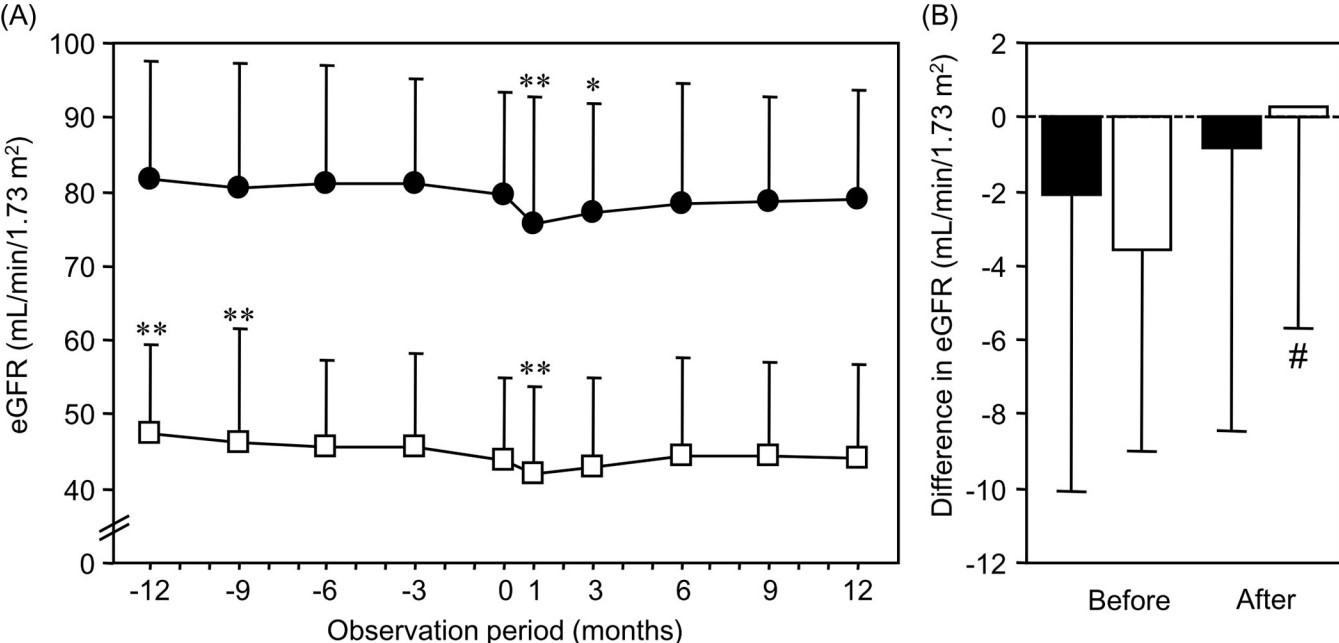

**Fig 2. Changes in the eGFR in the groups according to the eGFR at baseline (0 months) before and after the initiation of tofogliflozin (n = 102).** (A) The closed (black) circles and open (white) squares indicate subjects in the normal- ($\geq$60 mL/min/1.73 m$^2$, n = 68) and low- ($<$60 mL/min/1.73 m$^2$, n = 34) eGFR groups, respectively. *P $<$0.05 and **P $<$0.01 vs. baseline (0 months) value. (B) Closed (black) and open (white) bars indicate the differences in the eGFR in the normal- and low-eGFR group, respectively. #P $<$0.05 vs. corresponding value before the initiation of tofogliflozin. eGFR, estimated glomerular filtration rate.

### Changes in the eGFR in the groups according to the eGFR at baseline before and after the initiation of tofogliflozin

Fig 2 and Table 3 show the changes in the eGFR in 102 subjects who continued visiting over 12 months before the tofogliflozin administration. Although the eGFR gradually decreased before the initiation of tofogliflozin in the low-eGFR group (n = 34), it did not change significantly, except for the initial eGFR dip, after tofogliflozin administration. The change in the eGFR (0.2±6.0 mL/min/1.73 m$^2$) was significantly (P = 0.02) improved after tofogliflozin administration compared to before administration (-3.6±5.5 mL/min/1.73 m$^2$). No statistical sample-size calculations were conducted before the present retrospective study. However, we calculated the sample size required to compare the changes in the eGFR before and after the initiation of tofogliflozin for the subjects in the low-eGFR group. After setting the difference of the mean values and the common standard deviation at 3.8 and 6.0, respectively, the required sample size was calculated to be 29 (with a power of 90% and a 2-sided α level of 0.05). The eGFR did not significantly change before or after the initiation of tofogliflozin, except for the initial eGFR dip, in the normal-eGFR group (n = 68). Although there was no significant difference, the reduction in the eGFR was smaller after the initiation of tofogliflozin (-0.7±7.6 mL/min/1.73 m$^2$) than before (-2.0±8.2 mL/min/1.73 m$^2$).

### Relationships between the changes in the eGFR and clinical parameters in the FAS

Table 4 shows the relationships between the changes in the eGFR and the baseline clinical parameters in the FAS. In all subjects, the change in the eGFR showed a significant positive correlation with metformin use and a negative correlation with the HbA1c and eGFR at baseline (Fig 3A and 3B). In the normal-eGFR group, the change in the eGFR showed a significant

**Table 3. Changes in the eGFR in 102 patients treated over 12 months before the initiation of tofogliflozin administration.**

| eGFR | All subjects | Groups according to eGFR at baseline | |
|---|---|---|---|
| (mL/min/1.73 m$^2$) | | Normal-eGFR group | Low-eGFR group |
| | ($n$ = 102) | ($n$ = 68) | ($n$ = 34) |
| -12 months | 70.2±21.9** | 81.7±15.9 | 47.3±12.1** |
| -9 months | 69.1±22.8* | 80.1±16.7 | 46.3±15.1** |
| -6 months | 69.3±22.2* | 81.1±15.9 | 45.7±11.5 |
| -3 months | 69.2±21.6* | 81.0±14.2 | 45.6±12.5 |
| Baseline (0 months) | 67.7±21.3 | 79.6±13.8 | 43.7±11.1 |
| 1 month | 64.6±22.1** | 75.8±17.0** | 42.1±11.6** |
| 3 months | 65.8±21.4* | 77.3±14.6* | 42.8±12.2 |
| 6 months | 67.0±22.2 | 78.3±16.3 | 44.3±13.4 |
| 9 months | 67.4±21.1 | 78.8±14.1 | 44.5±12.6 |
| 12 months | 67.3±21.7 | 79.0±14.7 | 44.0±12.7 |
| Differences from baseline | | | |
| Before the initiation | -2.5±7.4 | -2.0±8.2 | -3.6±5.5 |
| After the initiation | -0.4±7.1 | -0.7±7.6 | 0.2±6.0# |

eGFR, estimated glomerular filtration rate.

* P<0.05

** P<0.01 vs. corresponding value at baseline.

# P<0.05 vs. corresponding value before the initiation of tofogliflozin.

negative correlation with the HbA1c and eGFR at baseline according to a multiple regression analysis with metformin use, HbA1c and eGFR included as independent variables. In the low-eGFR group, the change in the eGFR showed a significant negative correlation with only urate-lowering agent use. These statistical results did not change after any outliers, as assessed by boxplots, were excluded.

Table 5 shows the relationship between the changes in the eGFR and the changes in the clinical parameters considered to be related to the renal prognosis after the initiation of tofogliflozin in the FAS. The change in the eGFR showed a significant negative correlation with the change in the serum uric acid concentration according to a multiple regression analysis including the changes in HbA1c and uric acid concentrations as independent variables (Fig 3C and 3D). This significant negative correlation remained in both the normal- and low-eGFR groups.

## AEs in the SAS

In the SAS, tofogliflozin was discontinued in 17 patients (11%) at the onset of AEs (Fig 1). The details of AEs recorded during the observation period in the SAS are shown in Table 6. Overall AEs were recorded in 93 cases among 158 patients (59%) and were not significantly more frequent in the low-eGFR groups (61%) than in the normal-eGFR group (58%). The frequencies of individual AEs were also not significantly different between the normal- and low-eGFR groups. The most common AE was an increased urine volume in both the normal- and low-eGFR groups. Severe hypoglycemia was not observed in the present study.

## Discussion

In this study, tofogliflozin exerted renoprotection in the FAS, and this effect was obvious in the low-eGFR group. Although there was no statistically significant difference compared to the

**Table 4. Relationship between the changes in the eGFR and the clinical parameters at baseline in the full analysis set.**

| | All subjects (n = 130) | | | | Normal eGFR (n = 87) | | | | Low eGFR (n = 43) | |
|---|---|---|---|---|---|---|---|---|---|---|
| | Single regression | | Multiple regression | | Single regression | | Multiple regression | | Single regression | |
| | β | P | β | P | β | P | β | P | β | P |
| Male sex | 2.523 | 0.11 | | | 4.246 | 0.06 | | | -2.273 | 0.29 |
| Age (/years) | 0.003 | 0.96 | | | -0.009 | 0.91 | | | -0.085 | 0.37 |
| Duration of diabetes (/years) | 0.010 | 0.90 | | | -0.075 | 0.54 | | | 0.078 | 0.43 |
| Smoking history | 0.949 | 0.52 | | | 2.243 | 0.26 | | | -2.268 | 0.24 |
| Current drinker | -0.113 | 0.94 | | | -1.006 | 0.64 | | | 3.109 | 0.17 |
| Body weight (/kg) | 0.057 | 0.28 | | | 0.094 | 0.17 | | | -0.035 | 0.64 |
| Body mass index (/kg/m$^2$) | 0.075 | 0.63 | | | 0.010 | 0.62 | | | 0.005 | 0.98 |
| Systolic blood pressure (/mmHg) | -0.017 | 0.74 | | | -0.100 | 0.22 | | | 0.033 | 0.54 |
| Diastolic blood pressure (/mmHg) | -0.004 | 0.95 | | | -0.054 | 0.53 | | | 0.065 | 0.32 |
| Diabetic retinopathy | 1.239 | 0.49 | | | 0.826 | 0.75 | | | 1.132 | 0.60 |
| Diabetic peripheral neuropathy | 1.447 | 0.40 | | | 1.802 | 0.45 | | | 0.351 | 0.88 |
| Cerebrovascular disease | -0.667 | 0.73 | | | 0.295 | 0.91 | | | -2.680 | 0.24 |
| Coronary heart disease | 2.890 | 0.12 | | | 4.927 | 0.06 | | | -0.735 | 0.74 |
| Obesity | 0.419 | 0.79 | | | 1.443 | 0.50 | | | -2.516 | 0.24 |
| Hypertension | -0.233 | 0.90 | | | -1.064 | 0.79 | | | 0.875 | 0.79 |
| Albuminuria | -1.187 | 0.44 | | | -2.092 | 0.33 | | | -1.791 | 0.39 |
| Hyperuricemia | -0.660 | 0.67 | | | 0.447 | 0.85 | | | -3.049 | 0.10 |
| RAAS inhibitors use† | -0.263 | 0.86 | | | 0.263 | 0.89 | | | -2.639 | 0.17 |
| Calcium channel blockers use | 1.457 | 0.32 | | | 0.641 | 0.75 | | | 1.763 | 0.35 |
| Urate-lowering agents use | -1.167 | 0.38 | | | -0.933 | 0.73 | | | -4.032 | 0.03 |
| Metformin use | 2.976 | 0.04 | 3.188 | 0.02 | 4.417 | 0.03 | 2.317 | 0.22 | 1.470 | 0.43 |
| Sulfonylureas use | 5.185 | 0.08 | | | 7.451 | 0.07 | | | 1.100 | 0.76 |
| Thiazolidinediones use | -2.376 | 0.49 | | | -1.676 | 0.69 | | | -3.718 | 0.55 |
| α-glucosidase inhibitors use | 0.985 | 0.77 | | | 1.756 | 0.65 | | | | |
| Glinides use | -1.726 | 0.72 | | | -1.027 | 0.85 | | | | |
| DPP-4 inhibitors use | 0.581 | 0.69 | | | 0.126 | 0.95 | | | 0.731 | 0.70 |
| GLP-1 receptor agonists use | 2.001 | 0.37 | | | 3.248 | 0.29 | | | -0.497 | 0.86 |
| Insulin use | -0.623 | 0.73 | | | -1.451 | 0.60 | | | -0.679 | 0.74 |
| uPE (/mg/dL) | 0.011 | 0.25 | | | 0.022 | 0.28 | | | 0.002 | 0.86 |
| uACR (/mg/gCr) | 0.001 | 0.37 | | | 0.002 | 0.69 | | | 0.000 | 0.78 |
| Urinary L-FABP (/µg/gCr) | -0.011 | 0.79 | | | -0.010 | 0.84 | | | -0.059 | 0.50 |
| HbA1c (/%) | -1.430 | <0.01 | -1.148 | 0.01 | -2.179 | <0.01 | -1.640 | <0.01 | 0.462 | 0.49 |
| Hemoglobin (/g/L) | 0.298 | 0.55 | | | 0.643 | 0.35 | | | -0.251 | 0.68 |
| Uric acid (/µmol/L) | 0.017 | 0.09 | | | 0.012 | 0.39 | | | 0.027 | 0.05 |
| eGFR (/mL/min/1.73 m$^2$) | -0.102 | <0.01 | -0.101 | <0.01 | -0.219 | <0.01 | -0.174 | <0.01 | 0.035 | 0.70 |

eGFR, estimated glomerular filtration rate; uPE, urinary protein excretion; LDL, low-density lipoprotein; HDL, high-density lipoprotein; RAAS, renin-angiotensin-aldosterone system; DPP-4, dipeptidyl peptidase-4; GLP-1, glucagon-like peptide-1; uACR, urinary albumin-to-creatinine ratio; L-FABP, liver-type fatty acid-binding protein.

† Diabetic retinopathy includes simple, preproliferative and proliferative retinopathy.

‡ RAAS inhibitors include angiotensin-converting enzyme inhibitors, angiotensin II receptor blockers and aldosterone receptor antagonists.

change in eGFR in the normal-eGFR group, the recovery of eGFR in the low-eGFR group seems clinically important. A similar result was also observed in the subgroup that continued visiting over 12 months before the tofogliflozin administration.

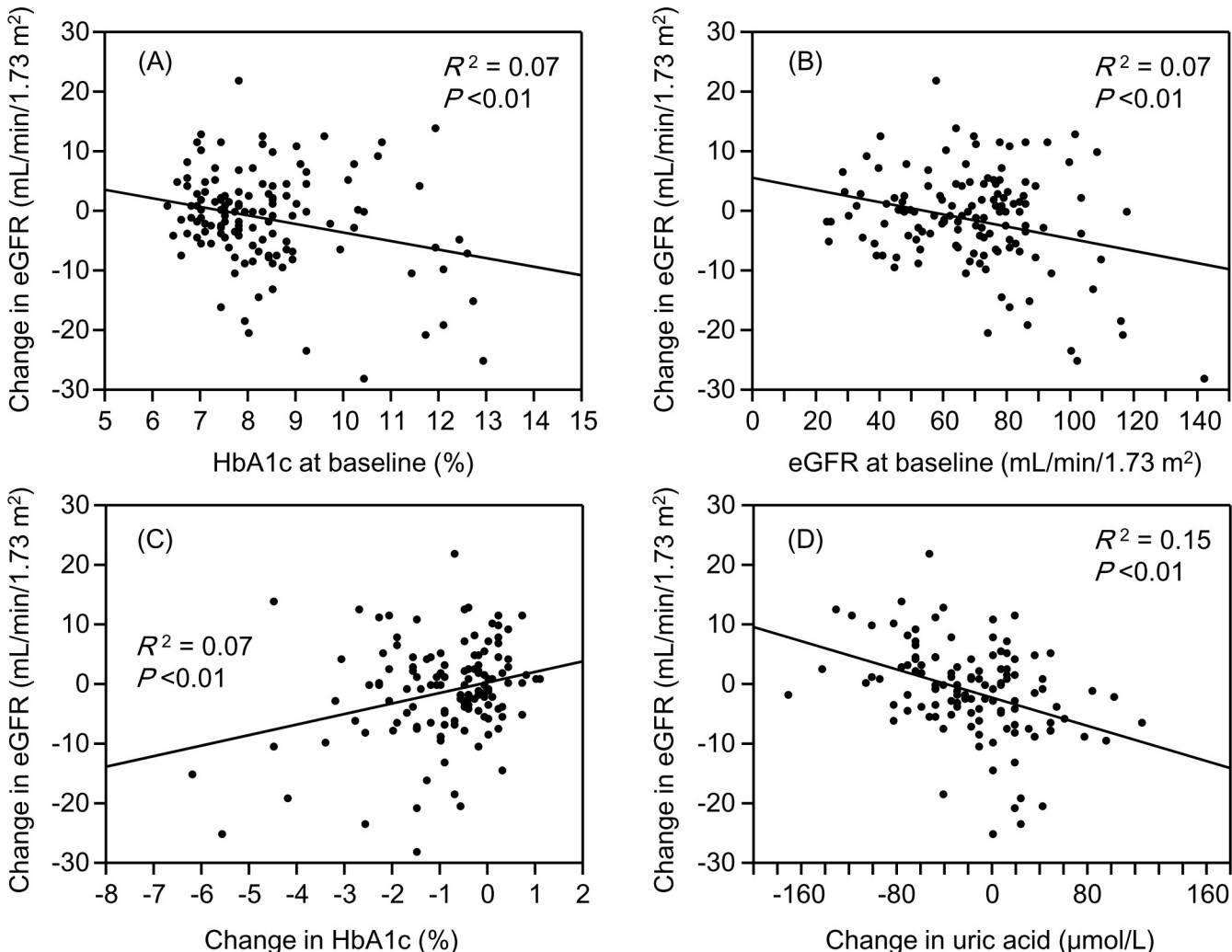

**Fig 3.** Relationships between the change in the eGFR after the initiation of tofogliflozin and (A) HbA1c at the baseline, (B) eGFR at the baseline, (C) change in HbA1c after the initiation of tofogliflozin and (D) change in serum uric acid concentration after the initiation of tofogliflozin. GFR, estimated glomerular filtration rate.

Similar to post-marketing surveillances [23–25], the reduction in HbA1c after starting tofogliflozin in the low-eGFR group was inferior to that in the normal-eGFR group. On the other hand, body weight, systolic blood pressure, uPE, uACR, urinary L-FABP, and serum uric acid decreased, and hemoglobin increased in the low-eGFR group. According to the statistical analysis, these changes were equal to or significantly greater than those in the normal-eGFR group and were considered to be clinically related to the long-term prognosis of renal function. It is well known that SGLT2 inhibitors reduce proteinuria secondary to a degree that not only corrects hyperglycemia but also decreases the intraglomerular pressure by restoring the damaged tubuloglomerular feedback in the kidneys [26–28]. These multifactorial improvements are considered to be why renal protection was observed in both the normal- and low-eGFR groups. Li and Gohda et al. showed the renoprotective effect of tofogliflozin in KK-Ay mice as an animal model of diabetic kidney disease [29]. Nagata et al. also reported that the administration of tofogliflozin reduced uACR and improved creatinine clearance in db/db mice with renal damage [30]. These studies demonstrated renoprotection by tofogliflozin based on

**Table 5. Relationship between the changes in the eGFR and the changes in the clinical parameters in the full analysis set.**

| | All subjects (n = 130) | | | | Normal eGFR (n = 87) | | | | Low eGFR (n = 43) | |
| | Single regression | | Multiple regression | | Single regression | | Multiple regression | | Single regression | |
| | β | P | β | P | β | P | β | P | β | P |
|---|---|---|---|---|---|---|---|---|---|---|
| Δ Body weight (/kg) | -0.160 | 0.49 | | | -0.232 | 0.41 | | | 0.213 | 0.65 |
| Δ Systolic blood pressure (/mmHg) | 0.042 | 0.43 | | | 0.088 | 0.25 | | | 0.001 | 0.99 |
| Δ Diastolic blood pressure (/mmHg) | 0.034 | 0.64 | | | 0.092 | 0.23 | | | -0.152 | 0.18 |
| Δ uPE (/mg/dL) | -0.001 | 0.47 | | | -0.031 | 0.29 | | | 0.004 | 0.75 |
| Δ uACR (/mg/gCr) | -0.001 | 0.81 | | | 0.005 | 0.73 | | | 0.001 | 0.64 |
| Δ Urinary L-FABP (/μg/gCr) | -0.007 | 0.95 | | | -0.016 | 0.94 | | | 0.052 | 0.68 |
| Δ HbA1c (/%) | 1.763 | <0.01 | 1.002 | 0.09 | 2.300 | <0.01 | 1.484 | 0.06 | -0.181 | 0.84 |
| Δ Hemoglobin (/g/L) | -0.981 | 0.20 | | | -0.811 | 0.43 | | | -1.073 | 0.30 |
| Δ Uric acid (/μmol/L) | -0.059 | <0.01 | -0.055 | <0.01 | -0.071 | <0.01 | -0.066 | <0.01 | -0.043 | 0.01 |

eGFR, estimated glomerular filtration rate; uPE, urinary protein excretion; uACR, urinary albumin-to-creatinine ratio; L-FABP, liver-type fatty acid-binding protein.

animal experiments. Sugiyama et al. reported the renoprotective effect of SGLT2 inhibitor therapy in patients with type 2 diabetes and renal dysfunction (CKD stages 3b-4) in a relatively small number of study subjects (n = 42), including 5 patients who received tofogliflozin [31]. Our study is the first report showing that tofogliflozin administration improved the renal prognosis associated with the changes in the clinical parameters predicting the renal prognosis in actual clinical practice.

Urinary L-FABP, a biomarker of tubulointerstitial injury of the kidneys, was also significantly reduced in the low-eGFR group of the current study, although similar results were also observed when another SGLT2 inhibitor was administered to patients with type 2 diabetes [11]. Whereas albuminuria reflects glomerular damage, urinary L-FABP levels increase when fatty acids overload the proximal tubules, such as ischemia and exposure to nephrotoxic

**Table 6. Adverse events during the observation period in the safety analysis set.**

| | All subjects | | Groups according to eGFR at baseline | | | |
| | | | Normal eGFR | | Low eGFR | |
| | (n = 158) | | (n = 101) | | (n = 57) | |
| | Total | Discontinuation | Total | Discontinuation | Total | Discontinuation |
|---|---|---|---|---|---|---|
| Urogenital infection | 5 (3%) | 3 (2%) | 4 (4%) | 2 (2%) | 1 (2%) | 1 (2%) |
| Severe hypoglycemia | 0 | 0 | 0 | 0 | 0 | 0 |
| Increased urine volume | 32 (20%) | 4 (3%) | 22 (22%) | 3 (3%) | 10 (18%) | 1 (2%) |
| Volume depletion | 6 (4%) | 1 (1%) | 4 (4%) | 1 (1%) | 2 (4%) | 0 |
| Cerebral infarction | 1 (1%) | 1 (1%) | 1 (1%) | 1 (1%) | 0 | 0 |
| Skin itching/eruption | 0 | 0 | 0 | 0 | 0 | 0 |
| Gastrointestinal symptoms† | 4 (3%) | 1 (1%) | 1 (1%) | 0 | 3 (5%) | 1 (2%) |
| Neoplasm | 7 (4%) | 1 (1%) | 2 (2%) | 0 | 4 (7%) | 1 (2%) |
| COVID-19 | 3 (2%) | 0 | 2 (2%) | 0 | 1 (2%) | 0 |
| Others | 34 (22%) | 5 (3%) | 22 (22%) | 4 (4%) | 12 (21%) | 1 (2%) |
| Death | 1 (1%) | 1 (1%) | 0 | 0 | 1 (2%) | 1 (2%) |
| Total | 93 (59%) | 17 (11%) | 58 (58%) | 11 (11%) | 35 (61%) | 6 (11%) |

eGFR, estimated glomerular filtration rate, COVID-19, Coronavirus disease 2019.

† Gastrointestinal symptoms include nausea, vomiting, abdominal pain, constipation and diarrhea.

substances [20]. In patients with chronic kidney disease, tubulointerstitial injury is more strongly linked to renal impairment than damage to the glomeruli. This is because the interstitium, which includes the renal tubules but not the glomeruli, makes up a significant portion of the kidney's anatomy [32]. It has been reported that tofogliflozin suppresses tubulointerstitial injury [33–35]. Shimomura et al. reported that urinary neutrophil gelatinase-associated lipocalin (NGAL), which is a tubular damage marker, tended to be decreased after the administration of tofogliflozin in 14 non-albuminuric patients with type 2 diabetes [33]. Nunoi et al. reported that both urinary N-acetyl-beta-d-glucosaminidase (NAG) and β2-microglobulin were decreased after the initiation of tofogliflozin, with significant negative correlations seen between changes in these markers and their corresponding baseline values in patients with type 2 diabetes and macroalbuminuria based on an integrated analysis of four phases 2 and 3 studies [34]. Furthermore, Ishibashi et al. showed that the monocyte chemoattractant protein-1 (MCP-1) gene expression and apoptotic cell death induced by exposure to high glucose levels were blocked by tofogliflozin in cultured human proximal tubular cells [35]. Therefore, the decrease in urinary L-FABP in this study may have been due to the reduction of ischemia and oxidative stress resulting from the workload being reduced by SGLT2 inhibition in the renal tubules [26–28]. Tofogliflozin may thus exert a protective mechanism for the interstitium of the kidney apart from effects on glomeruli.

Tofogliflozin is characterized by having the shortest half-life among SGLT2 inhibitors [13,14]. Therefore, there is concern that the antidiabetic effect may be inferior to those of other SGLT2 inhibitors; however, the induced improvement in blood glucose is reported comparable to that of other SGLT2 inhibitors [14,36]. If the renoprotective effect of tofogliflozin is also equivalent to that of other SGLT2 inhibitors, tofogliflozin will be a preferred option for patients with type 2 diabetes and renal impairment as tofogliflozin may preserve the patient's QOL by reducing the frequency of nocturnal urination [15,16], which is frequently seen in patients with diabetes.

In this study, the change in the eGFR was significantly associated with the eGFR and HbA1c at baseline in the normal-eGFR group. The relationship between the change in the eGFR and the eGFR at baseline is consistent with the results from previous studies [1–12]. Glomerular hyperfiltration is considered to be prominent when HbA1c is high in diabetic patients with preserved renal function [26–28]. Such a relationship between hyperglycemia and the eGFR appears to have remained even after adjusting for confounding factors by multivariate analysis. However, the change in the GFR was significantly associated with the use of urate-lowering agents in the low-eGFR group. Furthermore, there was a significant relationship between the change in the eGFR and the change in the uric acid concentration. Tubulointerstitial damage is considered to affect the renal prognosis in patients with hyperuricemia requiring drug therapy. Although hyperuricemia is a risk factor for the progression of renal dysfunction in patients with type 2 diabetes under SGLT2 inhibitor treatment [37], SGLT2 inhibitors have a decreasing effect on serum uric acid [38–42] due to an increase in urinary urate excretion through the upregulation of uric acid transporters of the renal tubules [43,44]. The reduction in the serum uric acid level in the present study was attenuated in the low-eGFR group, similar to previous reports [39–42]; however, the change in the eGFR was negatively associated with the change in the uric acid level. Although a reduced uric acid level induced by tofogliflozin may improve the prognosis by reducing tubulointerstitial damage, this effect is considered to be larger in patients with a preserved renal function than in those with renal insufficiency.

The reduction in uPE, uACR and urinary L-FABP was not necessarily associated with the change in the eGFR in the present study. This fact is likely due to the short observation period. Because the reduction in proteinuria partly reflects the improvement of glomerular hypertension, it appears to be renoprotective in the longer term.

Why the combination of tofogliflozin and metformin was associated with the change in the eGFR in the normal-eGFR group of the present study is unclear. The baseline eGFR among the metformin users (78.9±13.5 mL/min/1.73 m$^2$) was lower than among non-users (83.2 ±17.1 mL/min/1.73 m$^2$). Although it was not significant, this difference may have influenced the changes in the two groups.

The lower limit of the eGFR at which SGLT2 inhibitors are effective for renoprotection has not been clarified, although according to previous clinical trials [9,10], it was 25–30 mL/min/1.73 m$^2$. The minimum value of eGFR was 25 mL/min/1.73 m$^2$ in our previous study using luseogliflozin [11] and 23.1 mL/min/1.73 m$^2$ in the present study. Recently, the American Diabetes Association issued a recommendation that SGLT2 inhibitors should be used in patients with type 2 diabetes and an eGFR of ≥20 mL/min/1.73 m$^2$ [45,46]. Renin-angiotensin-aldosterone system inhibitors may occasionally be difficult to use due to a risk of hyperkalemia and a decline in the eGFR in diabetic patients with renal impairment. The availability of SGLT2 inhibitors may be good news for diabetic patients with low eGFR.

The frequencies of overall and individual AEs were not markedly different between the normal- and low-eGFR groups in the present study, although AEs associated with SGLT2 inhibitor use are generally more frequently observed in subjects with renal dysfunction than in those without it [11,23–25,47]. The prescription of tofogliflozin in this study was performed more than five years after SGLT2 inhibitors were first launched in Japan. Therefore, drug-specific AEs and the patients who were unsuitable for the use of SGLT2 inhibitors, such as those with cognitive impairment or frailty, had already been identified at the start of our study. Furthermore, patients were notified about the risk of AEs at the time of tofogliflozin prescription. These are considered to be reasons why AEs were not increased in the low-eGFR group compared to the normal-eGFR group and may suggest that tofogliflozin is safe enough to use even in patients with type 2 diabetes and renal impairment.

Several limitations should be mentioned regarding the current study. First, the present study was retrospectively performed in a relatively small number of diabetic patients. Sample sizes required to compare the changes in the eGFR between the normal- and low-eGFR groups were 259 (with a power of 90% and a 2-sided α level of 0.05). The statistical power was calculated to be 0.51 according to the subject's number of the FAS. Thus, it is necessary to pay attention to the possibility that changes in clinical parameters, such as body weight, blood pressure, proteinuria, hemoglobin, uric acid and eGFR, occurred incidentally. In addition, we did not examine the influence of medications other than the drugs excluded in the study setting (e.g., dose change of antidiabetic agents and short-term use of non-steroidal anti-inflammatory drugs and/or antidiabetics, etc). This fact may have impaired the accuracy of the assessment of the influence of tofogliflozin on renal function and blood glucose levels. It is necessary to conduct further prospective investigations on a larger number of patients to confirm our results. Furthermore, the selection of study subjects, especially in the low-eGFR group, might have been biased, as the attending physician might have administered tofogliflozin to patients who were unlikely to develop AEs. The study population showed a male predominance. The actual male/female ratio of Japanese patients with type 2 diabetes is unknown because Japan has no registration system. Our previous reports demonstrated that 60–62% of outpatients with type 2 diabetes were men [48–53], and the proportion in the present study (71%) appears to be higher than those reports. The proportions of male subjects were also 71–73% in our reports investigating the effects of SGLT2 inhibitors in patients with type 2 diabetes [11,12], which seems similar to the proportion in the present study. The present study did not establish criteria for the indication of tofogliflozin treatment because it was not a randomized investigation. A possible reason for this high ratio of male subjects may be the avoidance of patients who may develop genital urinary tract infections, an AE of SGLT2 inhibitor treatment. In this

study, 17 of 22 (77%) patients with cerebrovascular disease and 22 of 24 (92%) patients with coronary heart disease were men. In such cases, SGLT2 inhibitors, which have been reported to be effective for secondary prevention [1–10], might have been positively selected. It is considered that these circumstances combined to cause the difference in the male/female ratio between the present study and our previous investigations. Therefore, the subjects of this study did not reflect the general clinical characteristics of Japanese patients with type 2 diabetes. Second, the observation period in this study was short, before and after the initiation of tofogliflozin. Although a more extended observation was desirable, it is considered difficult to accurately evaluate the effects of tofogliflozin because drugs other than tofogliflozin are often changed in actual clinical practice. Third, the current study was unable to examine adherence to non-pharmacological therapy, such as calorie or salt restriction and pharmacological therapy. Because medication adherence is low in younger patients with type 2 diabetes [54], the outcome of our study may have been influenced by the patient's age in the normal-eGFR group. Fourth, the GFR was determined through the formula suggested by the Japanese Society of Nephrology [17] instead of using inulin clearance, the gold standard to evaluate the GFR. Because the eGFR was calculated based on the serum creatinine concentration, it should be noted that the renal function may have been overestimated in the elderly with a decreased skeletal muscle mass, who were frequently found in the low-eGFR group. It was reported that tofogliflozin administration, similarly to other SGLT2 inhibitors, reduced the skeletal muscle mass in patients with type 2 diabetes, but it was not as significant as the reduction in body fat [55]. Because a decrease in the serum creatinine concentration secondary to a reduced skeletal muscle mass after the starting of tofogliflozin may influence the preservation of the eGFR in elderly subjects in the low-eGFR group, the body composition should have been determined before and after the study.

However, despite these limitations, we believe that tofogliflozin is effective for protecting renal function and is clinically safe for patients with type 2 diabetes and renal dysfunction. Although at least part of the present study is similar to previously reported results, the observation of a renal protection effect of tofogliflozin without an increase in AEs is considered valuable as real-world data.

## Conclusion

Tofogliflozin may preserve renal function in the medium term in patients with type 2 diabetes and kidney impairment without an increase in specific AEs. Further studies are considered necessary because the current study was performed in a relatively small number of diabetic patients.

## Supporting information

**S1 Checklist. STROBE statement—checklist of items that should be included in reports of observational studies.**
(DOCX)

**S1 Dataset. Dataset for data analyses.**
(XLSX)

## Acknowledgments

The authors thank Tomoko Koyanagi in the secretarial section of Edogawa Hospital for her valuable help with data collection.

## Author Contributions

**Conceptualization:** Hiroyuki Ito.

**Data curation:** Hiroyuki Ito.

**Formal analysis:** Hiroyuki Ito.

**Funding acquisition:** Hiroyuki Ito.

**Investigation:** Hiroyuki Ito, Hideyuki Inoue, Takuma Izutsu, Suzuko Matsumoto, Shinichi Antoku, Tomoko Yamasaki, Toshiko Mori, Michiko Togane.

**Methodology:** Hiroyuki Ito.

**Project administration:** Hiroyuki Ito.

**Writing – original draft:** Hiroyuki Ito.

**Writing – review & editing:** Hiroyuki Ito, Hideyuki Inoue.

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
