## [Decision Letter · Decision Letter 0]

15 Jun 2023

PONE-D-23-06430Changes in the estimated glomerular filtration rate and predictors of the renal prognosis in Japanese patients with type 2 diabetes: A retrospective study during the 12 months after the initiation of tofogliflozinPLOS ONE

Dear Dr. Ito,

Thank you for submitting your manuscript to PLOS ONE. After careful consideration, we feel that it has merit but does not fully meet PLOS ONE’s publication criteria as it currently stands. Therefore, we invite you to submit a revised version of the manuscript that addresses the points raised during the review process.

We look forward to receiving your revised manuscript.

Kind regards,

Jaya Anna George, BMBCH

Academic Editor

PLOS ONE

Journal Requirements:

2. Thank you for submitting the above manuscript to PLOS ONE. During our internal evaluation of the manuscript, we found significant text overlap between your submission and previous work in the Results and Discussion sections of your manuscript. We would like to make you aware that copying extracts from previous publications, especially outside the methods section, word-for-word is unacceptable. In addition, the reproduction of text from published reports has implications for the copyright that may apply to the publications. Please revise the manuscript to rephrase the duplicated text, cite your sources, and provide details as to how the current manuscript advances on previous work. Please note that further consideration is dependent on the submission of a manuscript that addresses these concerns about the overlap in text with published work. We will carefully review your manuscript upon resubmission and further consideration of the manuscript is dependent on the text overlap being addressed in full. Please ensure that your revision is thorough as failure to address the concerns to our satisfaction may result in your submission not being considered further.

“This work was partly supported by Kowa Company, Ltd. (https://www.kowa.co.jp/). The funders had no role in study design, data collection and analysis, decision to publish, or preparation of the manuscript. There was no additional external funding received for this study.”

“H Ito has received funding support and lecture fees from Kowa Company, Ltd. and Taisho Pharmaceutical Co., Ltd., and lecture fees from Eli Lilly Japan KK, Novo Nordisk Pharma Ltd., Sumitomo Pharma Co., Ltd., Boehringer Ingelheim, Sanofi KK, Daiichi Sankyo Company, Novartis Pharma KK, Takeda Pharmaceutical Company Ltd., MSD KK, Astellas Pharma, Terumo Corporation, Mochida Pharmaceuticals, Teijin Pharma, Kissei Pharmaceuticals, Mitsubishi Tanabe Pharma Corporation, Sanwa Kagaku Kenkyusho, AstraZeneca KK, Kyowa Kirin Co. Ltd., Otsuka Pharmaceutical Co., Ltd., Bayer Yakuhin, Ltd., EA Pharma Co., Ltd., Ono Pharmaceutical Co., Ltd., Viatris Inc., and has received consulting fee from Becton, Dickinson and Company. H Inoue has received lecture fees from AstraZeneca KK. T Izutsu has received lecture fees from Boehringer Ingelheim, Novo Nordisk Pharma Ltd., Taisho Pharmaceutical Co., Ltd., Kowa Company, Ltd. and Asahi Kasei Pharma Corporation. S Matsumoto has received lecture fees from Eli Lilly Japan KK, Novo Nordisk Pharma Ltd., Astellas Pharma, Kyowa Kirin Co., Ltd. and AstraZeneca KK. S Antoku has received lecture fees from Kyowa Kirin Co. Ltd., Sanofi KK, Taisho Pharmaceutical Co., Ltd., Daiichi Sankyo Company, and Otsuka Pharmaceutical Co., Ltd. T Yamasaki, T Mori and M Togane have no conflict of interest.”

We note that one or more of the authors are employed by a commercial company

Additional Editor Comments (if provided):

Dear Dr Ito,

Thank you for your submission . In its current format this manuscript is not suitable for publication as a full manuscripts. Perhaps, at best, providing the authors can address issues raised, it could be a short research letter simply to show the impact of drug treatment on diabetes and renal parameters overall, and stratified by eGFR stage – without making any of the current claims regarding kidney protection.

In addition to the issues raised by the reviewer please address the following issues:

1. the lack of clarity regarding whether it was investigator initiated or pharma driven

2. the absence of an apriori statistical analysis plan to predict sample sizes needed

3. the retrospective analysis and small sample size

4. the short follow up time

5. the relevance of the comparison of a smaller group with 12 months follow up prior to commencing the drug

6. the erroneous conclusions drawn by the investigators - as per my comments

J A George

Reviewers' comments:

Reviewer's Responses to Questions

**Comments to the Author**

1. Is the manuscript technically sound, and do the data support the conclusions?

Reviewer #1: Yes

2. Has the statistical analysis been performed appropriately and rigorously? 

Reviewer #1: Yes

3. Have the authors made all data underlying the findings in their manuscript fully available?

Reviewer #1: Yes

4. Is the manuscript presented in an intelligible fashion and written in standard English?

Reviewer #1: Yes

5. Review Comments to the Author

Reviewer #1: The study investigated the renal protective effects of tofogliflozin over a 1-year period. The manuscript is well-written and has a good flow. This study adds vital data on the drug in human subjects as only animal studies have investigated the effect on eGFR to date. For a drug with a short half-life, it is important to demonstrate that it can achieve the goal of preserving renal function in patients with diabetes.

Comments to authors:

1. Line 38 – 39: authors mention a negative eGFR correlation with baseline eGFR and HbA1c. Does this mean if a patient had a baseline eGFR of 100, they were likely to have an eGFR below 100 at 12 months (negative correlation)? How does this translate to a ‘protective’ effect?

2. Authors to mention and clarify the male preponderance (71%); is this reflective of DM II demographic in Japan or due to clinical criteria for tofogliflozin treatment?

3. Table 2: month 3,6 and 9 do not show any significant trends and can be omitted from the table to allow space to put actual p-values instead of ** or #

4. Line 190 – 191: the authors state that weight and systolic blood pressure gradually decreased over 12 months in both groups. The over-all weight loss in both groups is 3kg over 12 months and SBP is 4 and 7 mmHg, which are both modest changes. Authors should probably state this actual weight loss and SBP change averages per group instead of a general description. The main changes occurred in the first month and then plateaued to the end i.e. not really a trend over time.

5. Line 198 – 199: authors mention “in uPE and urinary L-FABP were larger in the low-eGFR group than in the normal GFR group, there were no significant differences” Please clarify statistical significance versus clinical significance when discussing result findings, throughout the manuscript. Readers should get clarity on the potential clinical benefit of the drug based on whether the stipulated results show a clinically meaningful change; p-values can be <0.001 but with negligible actual mean differences. This must be clearly discussed for the important parameters including end-point eGFR, weight, SBP, uric acid findings. Second example: line 221 – 223 “change in the eGFR (0.2±6.0 mL/min/1.73 m2) was significantly 222 (P=0.02) improved after tofogliflozin administration compared to before administration 223 (-3.6±5.5 mL/min/1.73 m2)”.

6. Statistics: Which least squares method (Linear, Weighted, Robust or Nonlinear) was used, and why. Were there any outliers: if yes, how were they handled?

7. The short title does not indicate the central factor, which is the drug, tofogliflozin.

6. PLOS authors have the option to publish the peer review history of their article (what does this mean?). If published, this will include your full peer review and any attached files.

Reviewer #1: No

---

## [Author Response · Author response to Decision Letter 0]

4 Jul 2023

Responses to the academic editor

1. The lack of clarity regarding whether it was investigator initiated or pharma driven

Response: We added the text in the Method section.

2. The absence of an apriori statistical analysis plan to predict sample sizes needed

Response: The sample sizes required to compare the changes in eGFR between the normal- and low-eGFR groups were 259 in the present study. The statistical power was calculated to be 0.51 according to the subject’s number of the FAS. Therefore, the sample size was too small to compare the changes in eGFR between the two groups. We added the text showing the present study's low statistical power in the revised manuscript's limitations.

3. The retrospective analysis and small sample size

Response: Although these limitations were mentioned in the initial submission version, we amended the text with more emphasis in the revised manuscript.

4. The short follow up time

5. The relevance of the comparison of a smaller group with 12 months follow up prior to commencing the drug

Response: We added the text as another limitation of this study in the revised manuscript. Although a more extended observation was desirable, as the editor pointed out, it isn’t easy to accurately evaluate the effects of tofogliflozin because drugs other than tofogliflozin are often changed in actual clinical practice.

6. The erroneous conclusions drawn by the investigators - as per my comments

Response: We weakened the tone of the conclusion of the revised manuscript.

 

Responses to the reviewers

1. Line 38 – 39: authors mention a negative eGFR correlation with baseline eGFR and HbA1c. Does this mean if a patient had a baseline eGFR of 100, they were likely to have an eGFR below 100 at 12 months (negative correlation)? How does this translate to a ‘protective’ effect? 

Response: We amended the Conclusions of the Abstract after adding the actual value of the initial dip to the Results.

2. Authors to mention and clarify the male preponderance (71%); is this reflective of DM II demographic in Japan or due to clinical criteria for tofogliflozin treatment? 

Response: The actual male/female ratio of Japanese patients with type 2 diabetes is unknown because Japan has no registration system. Our previous cross-sectional reports (Nephrol Dial Transplant 25: 1161, 2010, J Diabates Investig 1: 273, 2010, BMC Nephrol 13: 48, 2012, Nephron Extra 3: 66, 2013, J Diabetes Complications 26: 286, 2012, Diabetes Care 37: e7, 2014, Geriatr Gerontol Int 17: 24, 2017, etc.) demonstrated that 60-62% of outpatients with type 2 diabetes were men, and the proportion in the present study (71%) appears to be higher in comparison to those reports. The proportions of men were also 71-73% in our reports that investigated the effects of SGLT2 inhibitors in patients with type 2 diabetes (Diabetes Metab Syndr Obes 12: 1783, 2019, PLoS One 16: e0248577, 2021), and it seems similar to that in the present study.

 No criteria for the indication of tofogliflozin treatment were established in the present study because it was not randomized. A possible reason for this high ratio of male subjects may be the avoidance of patients who may develop genital urinary tract infections, an adverse event of SGLT2 inhibitors. In this study, 17 of 22 (77%) patients with cerebrovascular disease and 22 of 24 (92%) patients with coronary heart disease were men. In such cases, SGLT2 inhibitors, which have been reported to be effective for secondary prevention, might have been positively selected. It is considered that these circumstances combined to cause the difference in the male/female ratio between the present study and our previous investigations.

 We included considerations regarding the male/female ratio in the limitation of the Discussion section and text showing that the subjects of this study did not reflect the general clinical characteristics of Japanese patients with type 2 diabetes.

3. Table 2: month 3,6 and 9 do not show any significant trends and can be omitted from the table to allow space to put actual p-values instead of ** or # 

Response: We reconstructed Table 2 as Appendix 1. As the reviewer pointed out in the following paragraph, the changes in the clinical parameters did not necessarily show the same course. For example, the blood pressure did not change significantly after 3 months, although the body weight and HbA1c gradually decreased up to 6 months. It doesn’t seem easy to understand these trends from Appendix 1. Therefore, we would like to keep this table without deleting the values at 3, 6, and 9 months. The P values for comparing the changes between the normal-eGFR and low-eGFR groups are shown in the text of the revised manuscript. In addition, the errors in the baseline body weight value were fixed.

4. Line 190 – 191: the authors state that weight and systolic blood pressure gradually decreased over 12 months in both groups. The over-all weight loss in both groups is 3kg over 12 months and SBP is 4 and 7 mmHg, which are both modest changes. Authors should probably state this actual weight loss and SBP change averages per group instead of a general description. The main changes occurred in the first month and then plateaued to the end i.e. not really a trend over time.

Response: We amended the text that the reviewer pointed out.

5. Line 198 – 199: authors mention “in uPE and urinary L-FABP were larger in the low-eGFR group than in the normal GFR group, there were no significant differences” Please clarify statistical significance versus clinical significance when discussing result findings, throughout the manuscript. Readers should get clarity on the potential clinical benefit of the drug based on whether the stipulated results show a clinically meaningful change; p-values can be <0.001 but with negligible actual mean differences. This must be clearly discussed for the important parameters including end-point eGFR, weight, SBP, uric acid findings. Second example: line 221 – 223 “change in the eGFR (0.2±6.0 mL/min/1.73 m2) was significantly 222 (P=0.02) improved after tofogliflozin administration compared to before administration 223 (-3.6±5.5 mL/min/1.73 m2)”. 

Response: We amended the first and second paragraphs in the Discussion section in accordance with the reviewer’s suggestion.

6. Statistics: Which least squares method (Linear, Weighted, Robust or Nonlinear) was used, and why. Were there any outliers: if yes, how were they handled? 

Response: The present study used the linear least square method without the exclusion of outliers after confirming that these values were not caused by transcription errors when they were transcribed from the medical records. Even when the outliers assessed by boxplots were excluded, the statistical results did not change, as shown in Appendix 2 (differences from Table 4 are shown in red).

We added the statistical method and results to the Methods and Results sections.

In addition, there was an error in the correlation coefficient between the change in eGFR and the uric acid level in the initial submission version, so 1.117 was changed to 0.017.

7. The short title does not indicate the central factor, which is the drug, tofogliflozin. 

Response: We amended the short title.

  

Appendix 1: Table 2 after excluding the values at 3, 6, and 9 months (P values added)

 Baseline 1 month P† 12 months P† Change from baseline P‡

Body weight (kg) 

 All subjects (n=91) 74.3±14.7 72.9±14.5 <0.01 71.5±14.7 <0.01 -2.9±3.5 

 Normal-eGFR (n=63) 73.8±15.3 72.5±15.3 <0.01 71.1±15.3 <0.01 -2.8±3.8 

 Low-eGFR (n=28) 75.4±13.3 73.9±12.8 <0.01 72.4±13.5 <0.01 -3.0±2.7 0.78

Systolic blood pressure (mmHg) 

 All subjects (n=122) 135±15 131±13 <0.01 131±14 <0.01 -5±14 

 Normal-eGFR (n=82) 134±12 130±12 0.02 130±13 0.01 -4±13 

 Low-eGFR (n=40) 139±18 133±15 0.03 132±16 0.01 -7±16 0.36

Diastolic blood pressure (mmHg) 

 All subjects (n=119) 80±13 78±11 0.09 79±12 0.40 -2±12 

 Normal-eGFR (n=80) 80±12 78±11 0.06 80±12 0.99 -0±11 

 Low-eGFR (n=39) 79±16 78±12 0.77 77±12 0.09 -2±9 0.32

uPE (mg/dL) 

 All subjects (n=121) 35.5±76.9 18.6±47.6 <0.01 11.8±37.1 <0.01 -23.6±58.7 

 Normal-eGFR (n=81) 20.5±49.2 9.3±26.3 <0.01 6.4±24.2 <0.01 -14.1±35.3 

 Low-eGFR (n=40) 65.8±108.8 37.5±70.8 <0.01 22.9±53.4 <0.01 -42.9±86.5 0.12

uACR (mg/gCr) 

 All subjects (n=101) 224.8±644.2 151.0±438.5 <0.01 -73.7±259.1 

 Normal-eGFR (n=71) 93.3±273.9 80.9±271.8 0.15 -12.0±80.8 

 Low-eGFR (n=30) 535.8±1052.5 316.9±666.6 <0.01 -218.8±429.8 <0.01

Urinary L-FABP (μg/gCr) 

 All subjects (n=58) 12.0±26.0 9.8±26.5 0.02 -2.1±9.1 

 Normal-eGFR (n=39) 9.7±28.9 9.1±31.2 0.28 -0.6±6.0 

 Low-eGFR (n=19) 16.6±18.9 11.3±12.6 0.03 -5.2±13.2 0.14

HbA1c (%) 

 All subjects (n=129) 8.4±1.5 7.8±1.2 <0.01 7.4±1.0 <0.01 -0.9±1.3 

 Normal-eGFR (n=86) 8.4±1.6 7.9±1.3 <0.01 7.3±0.9 <0.01 -1.0±1.3 

 Low-eGFR (n=43) 8.3±1.4 7.8±1.0 <0.01 7.6±1.1 <0.01 -0.7±1.1# 0.06

Hemoglobin (g/L) 

 All subjects (n=130) 143±15 145±15 <0.01 147±16 <0.01 4±9 

 Normal-eGFR (n=87) 144±14 146±15 <0.01 148±17 <0.01 4±10 

 Low-eGFR (n=43) 142±15 143±15 0.50 145±15 0.03 2±9 0.36

Uric acid (μmol/L) 

 All subjects (n=118) 305±70 295±70 0.02 286±63 <0.01 -18±50 

 Normal-eGFR (n=78) 297±71 289±75 0.12 276±65 <0.01 -21±49 

 Low-eGFR (n=40) 319±67 306±59 0.10 305±53 0.09 -13±54 0.65

eGFR (mL/min/1.73 m2) 

 All subjects (n=130) 68.7±21.8 65.3±21.9 <0.01 67.5±21.1 0.19 -1.2±8.2 

 Normal-eGFR (n=87) 80.6±15.0 76.3±16.9 <0.01 78.7±14.4 0.15 -1.9±9.0 

 Low-eGFR (n=43) 44.6±10.4 43.1±11.3 <0.01 44.8±12.3 0.76 0.2±6.0 0.44

eGFR, estimated glomerular filtration rate; uPE, urinary protein excretion; uACR, urinary albumin-to-creatinine ratio; L-FABP, liver-type fatty acid-binding protein 

† P: vs. corresponding value at baseline, ‡ P: vs. corresponding value in the normal-eGFR group.

 

Appendix 2; Table 4 after excluding outliers

 All subjects (n=130) Normal eGFR (n=87) Low eGFR (n=43)

 Single regression Multiple regression Single regression Multiple regression Single regression

 β P β P β P β P β P

Male sex 2.523 0.11 4.246 0.06 -2.273 0.29

Age (/years) 0.003 0.96 -0.009 0.91 -0.085 0.37

Duration of diabetes (/years) -0.098 0.31 -0.141 0.28 -0.065 0.61

Smoking history 0.949 0.52 2.243 0.26 -2.268 0.24

Current drinker -0.113 0.94 -1.006 0.64 3.109 0.17

Body weight (/kg) 0.068 0.22 0.115 0.13 -0.035 0.64

Body mass index (/kg/m2) 0.087 0.59 0.010 0.62 0.005 0.98

Systolic blood pressure (/mmHg) -0.057 0.30 -0.100 0.22 -0.024 0.70

Diastolic blood pressure (/mmHg) -0.032 0.61 -0.054 0.53 0.029 0.71

Diabetic retinopathy 1.239 0.49 0.826 0.75 1.132 0.60

Diabetic peripheral neuropathy 1.447 0.40 1.802 0.45 0.351 0.88

Cerebrovascular disease -0.667 0.73 0.295 0.91 -2.680 0.24

Coronary heart disease 2.890 0.12 4.927 0.06 -0.735 0.74

Obesity 0.419 0.79 1.443 0.50 -2.516 0.24

Hypertension -0.233 0.90 -1.064 0.79 0.875 0.79

Albuminuria -1.187 0.44 -2.092 0.33 -1.791 0.39

Hyperuricemia -0.660 0.67 0.447 0.85 -3.049 0.10

RAAS inhibitors use† -0.263 0.86 0.263 0.89 -2.639 0.17

Calcium channel blockers use 1.457 0.32 0.641 0.75 1.763 0.35

Urate-lowering agents use -1.167 0.38 -0.933 0.73 -4.032 0.03

Metformin use 2.976 0.04 3.188 0.02 4.417 0.03 2.317 0.22 1.470 0.43

Sulfonylureas use 5.185 0.08 7.451 0.07 1.100 0.76

Thiazolidinediones use -2.376 0.49 -1.676 0.69 -3.718 0.55

α-glucosidase inhibitors use 0.985 0.77 1.756 0.65 

Glinides use -1.726 0.72 -1.027 0.85 

DPP-4 inhibitors use 0.581 0.69 0.126 0.95 0.731 0.70

GLP-1 receptor agonists use 2.001 0.37 3.248 0.29 -0.497 0.86

Insulin use -0.623 0.73 -1.451 0.60 -0.679 0.74

uPE (/mg/dL) 0.054 0.21 0.089 0.20 -0.084 0.23

uACR (/mg/gCr) -0.002 0.94 -0.015 0.69 0.034 0.54

Urinary L-FABP (/μg/gCr) 0.167 0.64 0.145 0.77 -0.079 0.88

HbA1c (/%) -1.430 <0.01 -1.148 0.01 -2.179 <0.01 -1.640 <0.01 0.462 0.49

Hemoglobin (/g/L) 0.298 0.55 0.643 0.35 -0.251 0.68

Uric acid (/μmol/L) 0.017 0.09 0.012 0.39 0.027 0.05

eGFR (/mL/min/1.73 m2) -0.102 <0.01 -0.101 <0.01 -0.219 <0.01 -0.174 <0.01 0.035 0.70

---

## [Decision Letter · Decision Letter 1]

11 Sep 2023

Changes in the estimated glomerular filtration rate and predictors of the renal prognosis in Japanese patients with type 2 diabetes: A retrospective study during the 12 months after the initiation of tofogliflozin

PONE-D-23-06430R1

Dear Dr. Ito,

We’re pleased to inform you that your manuscript has been judged scientifically suitable for publication and will be formally accepted for publication once it meets all outstanding technical requirements.

Kind regards,

Keiko Hosohata, Ph.D.

Academic Editor

PLOS ONE

Reviewers' comments:

Reviewer's Responses to Questions

**Comments to the Author**

1. If the authors have adequately addressed your comments raised in a previous round of review and you feel that this manuscript is now acceptable for publication, you may indicate that here to bypass the “Comments to the Author” section, enter your conflict of interest statement in the “Confidential to Editor” section, and submit your "Accept" recommendation.

Reviewer #1: All comments have been addressed

Reviewer #2: All comments have been addressed

2. Is the manuscript technically sound, and do the data support the conclusions?

Reviewer #1: Yes

Reviewer #2: Yes

3. Has the statistical analysis been performed appropriately and rigorously? 

Reviewer #1: Yes

Reviewer #2: Yes

4. Have the authors made all data underlying the findings in their manuscript fully available?

Reviewer #1: Yes

Reviewer #2: Yes

5. Is the manuscript presented in an intelligible fashion and written in standard English?

Reviewer #1: Yes

Reviewer #2: Yes

6. Review Comments to the Author

Reviewer #1: Dear Authors

Thank you for addressing the comments on the first submission of this very important study.

I am happy that the comments were adequately addressed and any previous issues have been clarified.

All the best with your publication; I look forward to reading the published paper.

Reviewer #2: (No Response)

7. PLOS authors have the option to publish the peer review history of their article (what does this mean?). If published, this will include your full peer review and any attached files.

Reviewer #1: No

Reviewer #2: **Yes: **Mohsen Abbasi-Kangevari

---

## [Editor Report · Acceptance letter]

13 Sep 2023

PONE-D-23-06430R1 

Changes in the estimated glomerular filtration rate and predictors of the renal prognosis in Japanese patients with type 2 diabetes: A retrospective study during the 12 months after the initiation of tofogliflozin 

Dear Dr. Ito:

I'm pleased to inform you that your manuscript has been deemed suitable for publication in PLOS ONE. Congratulations! Your manuscript is now with our production department. 

Kind regards, 

on behalf of

Dr Keiko Hosohata 

Academic Editor

PLOS ONE